# Do the Temperamental Characteristics of Both Mother and Child Influence the Well-Being of Adopted and Non-Adopted Children?

**DOI:** 10.3390/children9081227

**Published:** 2022-08-13

**Authors:** Krista Liskola, Hanna Raaska, Christian Hakulinen, Helena Lapinleimu, Marko Elovainio

**Affiliations:** 1Department of Child Psychiatry, Helsinki University Hospital, University of Helsinki, 00029 Helsinki, Finland; 2Department of Psychology and Logopedics, University of Helsinki, 00014 Helsinki, Finland; 3Department of Paediatrics and Adolescent Medicine, Turku University Hospital, University of Turku, 20521 Turku, Finland; 4Finnish Institute for Health and Welfare, 00271 Helsinki, Finland

**Keywords:** temperament, internationally adopted children, maternal distress, child behavioral problems

## Abstract

(1) Background: For decades, the temperaments of infants and small children have been a focus of studies in human development and been seen as a potential contributor to children’s developmental patterns. However, less is known about the interplay between the temperamental characteristics of mothers and their children in the context of explaining variations in developmental outcomes. The aim of our study was to explore the associations—with or without genetic links—of the temperaments and psychological distress of mothers and the temperaments of children with behavioral problems in a group of internationally adopted children and their adoptive mothers and in a group of non-adopted children and their mothers. (2) Methods: Data (*n* = 170) were derived from the ongoing Finnish Adoption (FinAdo) follow-up study. The children included were under the age of 7 years; 74 were adopted internationally through legal agencies between October 2010 and December 2016, and the remaining 96 were non-adopted children living with their birth parents (biological group) recruited from day-care centers. We used Mary Rothbart’s temperament questionnaires to assess temperament, the Child Behavior Checklist (CBCL) to obtain data on the children’s behavioral/emotional problems and competencies, and the General Health Questionnaire (GHQ) to assess parental psychological distress. The study was approved by the Ethics Committee of the Hospital District of Southwest Finland, and written informed consent was obtained from the parents and the children themselves. (3) Results: The negative affectivities of both mothers and children were associated with the total CBCL and with both internalizing and externalizing problem behaviors after adjusting for age, gender, and adoption status. Both relationships remained significant when tested simultaneously, suggesting additive effects. Maternal negative affect was associated with problem behavior irrespective of child extraversion/surgency. Child extraversion/surgency was associated with lower levels of all internalizing behavioral problems when adjusted for maternal sociability. Child negative affect was associated with all behavioral problem measures irrespective of maternal sociability or maternal psychological distress. Maternal distress was associated with child problem behaviors only in children with low extraversion/surgency. (4) Limitations: The sample size was relatively small, and the information was gathered solely with questionnaires. (5) Conclusions: The results of the study may be clinically significant. Child negative affect, maternal negative affect, and maternal experienced distress, combined with low child extraversion/surgency, may increase the risk of child problem behaviors in both adoptees and non-adoptees.

## 1. Introduction

Temperament has been described as individual differences in reactivity and self-regulation that are constitutionally based, appear early, and are influenced over time by heredity, maturation, and experience [1]. More recent research offers an alternative definition: “Temperament traits are early emerging basic dispositions in the domains of activity, affect, attention, and self-regulation, and these dispositions are the product of complex interactions among genetic, biological, and environmental factors across time” [2].

For decades, temperament in early life has been a focus of studies in human development and psychopathology in children and adolescents [3]. Individual differences in temperament reflect individual differences in central nervous system functioning and structure, and while there is a consistent body of evidence documenting the contributions of genotype to these differences, it has become increasingly clear that bioecological influences (e.g., nutrition) and environmental factors involving parenting styles and the nature of the home context can impose transgenerational effects [4,5].

Intergenerational transmission, a process through which an earlier generation influences parenting attitudes and behavior, genetically or otherwise, inevitably results in parent–offspring resemblance [6,7]. Although resemblance has been studied extensively in many areas of human functioning, studies into parent–offspring resemblance in personality are underrepresented [8], and resemblance has rarely been linked to underlying transmission mechanisms other than genetics [9,10]. 

Previous research has shown associations between child temperament, parenting, and child adjustment [11,12,13], and moreover, parental temperament traits have been found to predict especially externalizing problems in children [14,15,16]. In a recent study, Xing et al. assessed the combined effects of maternal personality and child temperamental reactivity on child externalizing behavioral problems [17]. In addition, disrupted attachment patterns and mismatches between behavioral expectations and the demands of peers or parents have been hypothesized to mediate the associations between child temperament and adjustment [12,18,19,20,21]. 

The mental well-being of adopted children has been addressed in previous studies; most adopted children are emotionally and behaviorally healthy but more likely than non-adoptees to receive mental health treatment. According to several studies, adopted children are referred for psychiatric treatment at a higher rate than their non-adopted peers [22,23], and in a meta-analysis of 11 studies, Askeland et al. [24] concluded that adopted adolescents experience more mental health problems. 

The aim of our study was to explore the associations between child behavioral problems, the temperaments and psychological distress of mothers, and the temperaments of children. The study’s design, which included adoptees and biological children, enabled us to examine whether these associations depend on genetic links between mothers and children. We decided to focus on two temperamental characteristics, negative affect and extraversion/surgency, as these characteristics have been consistently associated with later psychological problems [25].

## 2. Methods

### 2.1. Participants

This study was part of the ongoing Finnish Adoption (FinAdo) study [26,27], which is a unique and extensive Finnish medical study of internationally adopted children. The aim is to produce new, nationally, and internationally significant information on the physical and mental health and psychosocial development of these children. This kind of study offers a distinctive view on the development of psychiatric and developmental disorders because an adopted child’s rearing environment is transformed, often drastically, after their early years, and the parents and children have no shared genes. The FinAdo 2 project is an ongoing, prospective clinical follow-up study, and invitations to participate were sent through the three legalized adoption agencies working in Finland to all families who had internationally adopted a child under the age of 7 years between October 2010 and December 2016. The non-adopted children were recruited from day-care centers in the cities of Turku and Kaarina in southwest Finland. This temperament sub-study was conducted using questionnaires that were completed by the adoptive and biological parents. Of the 170 children, 74 were adopted and 96 non-adopted; 98 were boys and 72 girls; and the mean age was 4.17 years (SD 1.62). 

The current study was approved by the Ethics committee of the Hospital District of Southwest Finland, and written informed consent was obtained from the parents. 

### 2.2. Measures

#### 2.2.1. Temperament Characteristics

To assess child temperament, we used Mary Rothbart’s temperament questionnaires, specifically the Early Childhood Behavior Questionnaire (ECBQ) and the Children’s Behavior Questionnaire (CBQ) [28].

The ECBQ is a parent-report measure of temperament in children aged from 18 months to 3 years that assesses 18 dimensions of temperament (activity level/energy, attentional focusing, attentional shifting, cuddliness, discomfort, fear, frustration, high-intensity pleasure, impulsivity, inhibitory control, low-intensity pleasure, motor activation, perceptual sensitivity, positive anticipation, sadness, shyness, sociability, and soothability, which form three broad factors: surgency/extraversion, negative affect, and effortful control. A study by Putnam et al. [29] supports the reliability and validity of the ECBQ as a measure of varied and finely differentiated aspects of toddler temperament; the 18 scales were found to be internally consistent, and for most dimensions, different raters were consistent both with one another and across time. In our sample, the Cronbach’s’s alphas were 0.83 for negative affect and 0.79 for surgency/extraversion.

The CBQ was designed to measure temperament in children aged from 3 to 7 years and assesses 15 dimensions of temperament (activity level, anger/frustration, approach, attentional focusing, discomfort, falling reactivity and soothability, fear, high-intensity pleasure, impulsivity, inhibitory control, low-intensity pleasure, perceptual sensitivity, sadness, shyness, and smiling and laughter), which reliably form the same three factors as the ECBQ. Rothbart et al. [30] assessed the reliability and validity of the CBQ and concluded that its scales have adequate internal consistency and can be used in studies requiring a highly differentiated yet integrated measure of temperament. In our sample, the Cronbach’s’s alpha was 0.88 for negative affect and 0.91 for surgency/extraversion.

Maternal temperament was measured using the Emotionality–Activity–Sociability Temperament Survey [31] The questionnaire consists of 27 items on a 5-point scale of (1) totally disagree to (5) totally agree. Negative emotionality includes two components, anger and fear, which were measured with 12 items. Sociability was measured with five items. The Cronbach’s’s alphas were 0.81 for sociability and 0.78 for negative emotionality. Good psychometric properties have been reported in other studies [32,33].

#### 2.2.2. Parental Psychological Distress

The General Health Questionnaire (GHQ) is a self-administered screening questionnaire designed for use in consulting settings to detect individuals with diagnosable psychiatric disorders [34]. The most extensively used screening instrument for common mental disorders is the 12-item GHQ (GHQ-12). Various versions of the GHQ-12 have been reported to be useful in determining the presence of depression and to have good constructive validity [35]. The Cronbach’s alpha for our sample was 0.87.

#### 2.2.3. Behavioral Problems

To assess child behavioral problems, we used two CBCL questionnaires: the CBCL for ages 18 months to 5 years, which has 100 questions, and the CBCL for ages 6 to 18 years, which has 113 questions, excluding 5 open-ended questions. These questionnaires produce scores on the same dimensions and are thus comparable and age-adjusted. Each question is answered as (0) not true, (1) somewhat or sometimes true, or (2) very true or often true. The CBCL provides a total score for behavioral characteristics and separate scores for internalizing and externalizing behavioral symptoms. The total CBCL score was calculated by summing up all items. The higher a child’s scores on the CBCL, the more behavioral problems the child has [36]. The Cronbach’s alphas in our sample were as follows: CBCL total score, 0.93; internalizing problems, 0.82; externalizing problems, 0.88.

#### 2.2.4. Child-Related Background Factors

The child-related variables included the child’s adoption status (internationally adopted or not adopted), gender, age at adoption (if applicable), and age at the date of completing the questionnaire.

#### 2.2.5. Statistical Analyses

Differences in child and maternal temperamental characteristics, child problem behaviors, and maternal distress according to the adoption status were examined using Student’s *t*-test. The associations of child problem behaviors with maternal temperamental characteristics, maternal distress, and child temperamental characteristics were examined using linear regression analyses. Maternal temperamental characteristics, maternal distress, and child temperamental characteristics were inserted into the same regression model to test the possible additive nature of the associations. In a separate model, an interaction term between maternal temperamental characteristics or distress and child temperamental characteristics was inserted to examine possible interaction effects. In all analyses, adoption status, gender, and age were used as covariates. The associations are presented as regression coefficients (b) and standard errors (se). All analyses were performed using Stata 17.0. Mediation analyses were conducted using structural equation modeling and Medsem package [37].

## 3. Results

Descriptive statistics for the study sample are shown in Table 1. 

There were no significant differences in the temperamental dimensions between adopted and non-adopted children. Mothers of adopted children reported more externalizing behaviors than mothers of non-adopted children. The effect sizes ranged from small (0.02) to moderate (0.49). These are shown in Table 2. 

The coefficients of the correlations between maternal temperament, child temperament, and child problem behaviors are shown in Table 3. Child negative affect was associated with higher levels of behavioral problems, and there was a negative correlation between child extraversion/surgency and child internalizing symptoms. Higher maternal negative affect was associated with higher levels of maternal psychological distress.

The associations of maternal negative affect and child temperament characteristics with child problem behavior are shown in Table 4. Maternal and child negative affect were associated with the total CBCL, internalizing problem behaviors, and externalizing problem behaviors, and the relationships with total and external problem behaviors remained significant when tested simultaneously, suggesting an additive effect. In the model in which the relationships of child problem behaviors with maternal negative affect and child extraversion/surgency were examined, higher maternal negative affect was again associated with higher total, internal, and external problem behaviors. Higher child extraversion/surgency was associated with lower internal, but not total or external, problem behaviors. Maternal negative affect was associated with all child problem behavior measurements irrespective of child extraversion/surgency. No interaction effects in the association of maternal negative affect and child temperament characteristics with child problem behaviors were found.

The simultaneous associations of child problem behaviors with maternal sociability and child temperament characteristics are shown in Table 5. Whereas maternal and child extraversion/surgency were not associated with the total CBCL, higher child extraversion/surgency was associated with lower internal problem behaviors, and higher maternal sociability was associated with lower child external problem behaviors. Higher maternal extraversion/surgency was associated with lower total and externalizing problem behaviors irrespective of child negative affect. Child negative affect was associated with all problem behavior measures irrespective of maternal sociability. No interaction effects between maternal sociability and child temperament characteristics were found.

The associations of child problem behaviors with maternal psychological distress and child temperament characteristics are shown in Table 6. Maternal distress was associated with all problem behavior measures irrespective of child extraversion/surgency, and child extraversion/surgency was associated with lower internalizing problems even when adjusted for maternal distress. Child negative affect was associated with all problem behavior measures irrespective of maternal distress, but in contrast, there was no association between maternal psychological distress and child problem behaviors when examined together with child negative affect. An interaction effect suggesting a cross-over interaction of child problem behavior with maternal psychological distress and child surgency was found (Figure 1), and maternal distress was thus associated with problem behavior only in children with low surgency.

Last, in a multivariate model, the associations between child and maternal temperamental characteristics and maternal distress with child problem behaviors were examined (Table 7). The results of these analyses showed that child negative affect was consistently associated with child problem behaviors. The interaction term between adoption status and the independent variables (child and maternal temperamental characteristics and maternal distress) was not statistically significant (*p*-values > 0.05) suggesting that the adoption status did not modify the associations of child and maternal temperamental characteristics and maternal distress with child problem behaviors.

As sensitivity analyses, we tested whether child temperament traits would mediate the associations between maternal traits and child behavioral problems. Child negative affect mediated (*z* = 2.95, *p* < 0.01, 26% of total effect mediated) the association between maternal negative affect and child problem behavior (total score). None of the other mediation effects tested were statistically significant.

## 4. Discussion

The aim of our study was to explore the associations of child behavioral problems with the temperament and psychological distress of mothers and the temperament of children and to examine whether these associations are dependent on genetic links between mothers and children.

The results indicate that child temperamental traits are associated with behavioral problems in children, and controlling for adoption status did not change the results of the analyses, suggesting that the relationships apply equally to adopted children In univariable analyses, maternal negative affect and maternal distress also were associated with child problem behavior irrespective of child extraversion/surgency. However, in multivariable analyses, maternal psychological distress was associated with child problem behaviors only in children with low extraversion/surgency. In multivariable analyses, maternal lower sociability was associated with child externalizing behavioral problems, but maternal negative affect was not associated with child problem behavior. Low child extraversion/surgency was associated with internalizing behavioral problems Altogether, child negative affect, maternal low sociability, and low child extraversion/surgency may increase the risk of child problem behaviors in both adoptees and non-adoptees. Our sensitivity analyses suggested, however, that maternal negative affect may have an indirect effect on child problem behaviors through child negative affect.

Parents have an extensive influence on the children they raise. Parenting is culturally influenced, and therefore cultural differences in temperament can be anticipated [38], and in this regard, our study design is interesting: we could test whether the associations were affected by adoption status and thus explained by a shared genetic background. As stated earlier, individual differences in personality development have been shown to be predictive of the risk of and resilience to psychopathology [11,12,13], and Lunansky et al. [39] have proposed a personality–resilience–psychopathology model. Previous studies have shown that negative emotionality, effortful control, and surgency-related traits, such as impulsivity and behavioral inhibition, may contribute to the manifestation of both internalizing and externalizing problems in childhood [40,41,42,43]. The results of our study suggested the same, and moreover, the trait of extraversion/surgency seems to have a protective function. 

Adoptees have often been exposed to childhood trauma, such as neglect, family violence, physical and sexual abuse, placement instability, and institutionalized care, which increase the risk for attachment difficulties, posttraumatic stress disorder, and other psychiatric disorders [24,44]. These early experiences of deprivation are strong predictors of later psychopathology [45,46], and it could therefore be argued that temperamental differences play a less significant role for adopted children, but our study suggests that the temperamental characteristics of adopted children and their adoptive mothers do influence the children’s mental health.

Numerous studies have examined personality similarities between peers and romantic partners (e.g., [47]), but little is known regarding personality similarities between parents and offspring [48], and few previous studies have examined those similarities and their role in child development [4,13,49]. Franken et al. [48] investigated the impact of parent–offspring personality similarities on adolescent externalizing problems and found that child and parent personalities equally impact the development of externalizing problems during late adolescence. Two earlier studies [4,49] suggested that parent and offspring personality similarities might play an important role in child development, and both studies indicated that when similarity was a significant factor, it was associated with fewer externalizing problems. n a study by Rettew et al. [13], who examined the goodness-of-fit hypothesis regarding child–parent temperament and psychopathology, the interactions of the child and parent temperament dimensions were noteworthy additional predictors of childhood psychopathology. The interactions between the child and parent temperament dimensions predicted higher levels of externalizing, internalizing, and attention problems beyond the effects of these dimensions alone, and the authors concluded that the effect of child temperament upon child psychopathology can be dependent on parent temperament and that the effect of parent temperament can be dependent on child temperament [13]. The results of our sensitivity analyses indicated an indirect effect of maternal negative affect on child problem behavior through child negative affect and thus support previous studies. Furthermore, we concluded that maternal lower sociability was associated with child externalizing behavior.

The fact that temperamental differences affect psychological development irrespective of a genetic link is of importance to adoptive parents and their children, and the results of our study can thus be considered clinically significant.

### Strengths and Weaknesses

The major limitation of our study is the relatively small sample size (*N* = 170). An additional limitation is data collection using questionnaires, which meant that the assessments of behavioral problems, child temperament, maternal temperament, and maternal psychological distress were based solely on maternal reports. Furthermore, as the vast majority of the respondents were mothers, we were unable to assess any potential differences between mothers’ and fathers’ reports. Further studies would benefit from integrating the perspectives of multiple informants (e.g., fathers, teachers).

The major strengths of our study are the use of well-established and widely used measures (CBCL, GHQ, ECBQ, CBQ, and EAS) and the ability to consider the genetic links between parents and children. To our knowledge, this is the first study to assess the associations of child and maternal temperament with psychological well-being in both adoptive and non-adoptive families.

## 5. Conclusions

Our study suggests that certain temperamental features of children and mothers may impact the psychological development and well-being of children. Because adopted children are at a heightened risk of mental health problems, it is important to acknowledge any factors that may affect their psychological development. 

The results of our study help to identify those children who are at risk to develop behavioral problems. The risk is increased when maternal psychological distress is combined with low extraversion/surgency in the child and in the case of maternal and child negative affect, especially when mother and child share this trait. We are unable to change a child’s temperament, but when there is a risk of externalizing behavior, early interventions, such as Incredible Years, are available. These interventions have been studied and shown to be effective [50]. 

## Figures and Tables

**Figure 1 children-09-01227-f001:**
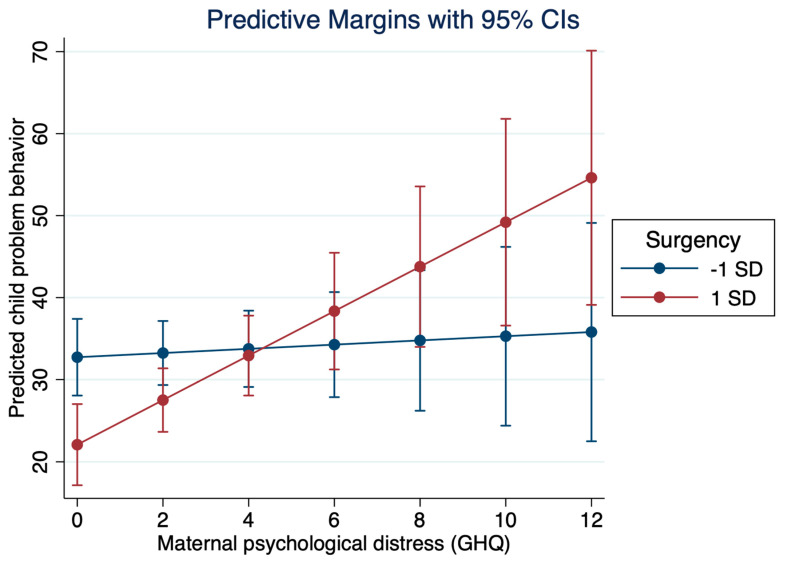
Interaction of child problem behavior and maternal psychological distress.

**Table 1 children-09-01227-t001:** Descriptive statistics.

	Mean	SD	N (%)
Children			170 (100)
Adopted Boy/Girl			74 (43.5) 98 (57.6)/72 (42.4)
Age	4.17	1.62	
CBCL Total	30.7	18.4	
Internal	7.41	5.62	
External	11.4	7.18	
Temperament			
Negative affect	3.42	0.63	
Extraversion/Surgency	4.77	0.63	
Mothers			
Temperament			
Negative affect	2.47	0.55	
Sociability	3.57	0.76	
Psychological distress	2.2	2.9	

N = Number, SD = standard deviation.

**Table 2 children-09-01227-t002:** Comparison of descriptive statistics according to adoption status.

	Adopted						
	No		Yes				
	Mean	SD	Mean	SD	Student’s T	*p*	Cohen’s D
Child negative affect	3.50	0.60	3.32	0.66	0.25	0.06	0.04
Child surgency/extraversion	4.70	0.63	4.85	0.63	−0.12	0.14	0.02
Maternal negative affect	2.48	0.58	2.46	0.50	1.86	0.80	0.29
Maternal sociability	3.56	0.79	3.58	0.72	−1.47	0.90	0.23
CBCL total	27.55	18.35	35.20	18.12	−2.71	0.01	0.42
CBCL internal	6.80	5.39	8.31	5.87	−1.75	0.08	0.27
CBCL external	9.93	6.92	13.35	7.06	−3.17	<0.01	0.49
GHQ	2.17	2.90	2.23	2.92	−0.15	0.88	0.02

**Table 3 children-09-01227-t003:** Correlation coefficients between the dependent and independent study variables.

	1	2	3	4	5	6	7	8
1. Mother: Negative affect	1	−0.20 **	0.31 **	−0.21 **	0.36 **	0.31 **	0.26 **	0.40 **
2. Mother: Sociability	−0.20 **	1	−0.03	0.17 *	−0.12	−0.09	−0.15	−0.15
3. Child: Negative affect	0.31 **	−0.03	1	−0.31 **	0.37 **	0.38	0.29	0.11
4. Child: Extraversion/Surgency	−0.21 **	0.17 *	−0.31 **		−0.14	−0.27	0.04	0.01
5. CBCL total	0.36 **	−0.12	0.37 **	−0.14	1	0.85 **	0.89 **	0.19 *
6. CBCL internal	0.31 **	−0.09	0.38 **	−0.27 **	0.85 **	1	0.67 **	0.15
7. CBCL external	0.26 **	−0.15	0.29 **	0.04	0.89 **	0.67 **	1	0.17 *
8. Maternal distress	0.40 **	−0.15	0.11	0.01	0.19 *	0.15	0.17 *	1

** *p* < 0.01, * *p* < 0.05. CBCL: Child Behavior Checklist [37].

**Table 4 children-09-01227-t004:** The associations between maternal and child negative affect and child problem behavior.

	CBCL Total				CBCL Internal				CBCL External			
	**b**	**se**	**b**	**se**	**b**	**se**	**b**	**se**	**b**	**se**	**b**	**se**
Mother: Negative affect	7.94 **	2.44	30.15 *	12.83	2.28 **	0.77	6.9	4.05	2.04 *	0.96	11.12 *	5.02
Child: Negative affect	11.47 **	2.30	26.67 **	8.92	3.15 **	0.72	6.31 *	2.82	3.94 **	0.9	10.15 **	3.49
Mother: Negative affect X Child: Negative affect			−6.33	3.59			−1.32	1.13			−2.59	1.4
R-squared	0.30		0.32		0.25		0.26		0.28		0.30	
	**b**	**se**	**b**	**se**	**b**	**se**	**b**	**se**	**b**	**se**	**b**	**se**
Mother: Negative affect	11.79 **	2.50	3.25	16.48	2.98 **	0.75	1.91	4.98	3.84 **	0.96	−3.33	6.34
Child: Extraversion/Surgency	−2.62	2.27	−7.13	8.91	−2.07 **	0.69	−2.64	2.69	0.88	0.88	−2.91	3.43
Mother: Negative affect X Child: Extraversion/Surgency			1.83	3.50			0.23	1.06			1.54	1.35
R-squared	0.20		0.20		0.20		0.21		0.20		0.21	

The associations are adjusted for age, gender, and adoption status. Values are regression coefficients (b) and standard errors (se). ** *p* < 0.01, * *p* < 0.05. CBCL: Child Behavior Checklist [37].

**Table 5 children-09-01227-t005:** The associations between maternal and child extraversion/surgency with child problem behavior.

	CBCL Total				CBCL Internal				CBCL External			
	**b**	**se**	**b**	**se**	**b**	**se**	**b**	**se**	**b**	**se**	**b**	**se**
Mother: Sociability	−2.50	1.89	−11.56	13.71	−0.39	0.56	−1.87	4.08	−1.62 *	0.71	0.57	5.15
Child: Extraversion/Surgency	−4.66	2.38	−11.58	10.63	−2.64 **	0.71	−3.77	3.16	0.39	0.89	2.06	3.99
Mother: Sociability X Child: Extraversion/Surgency			1.93	2.89			0.31	0.86			−0.47	1.08
R-squared	0.10		0.10		0.13		0.14		0.15		0.15	
	**b**	**se**	**b**	**se**	**b**	**se**	**b**	**se**	**b**	**se**	**b**	**se**
Mother: Sociability	−3.06	1.68	0.14	8.98	−0.73	0.53	0.72	2.82	−1.54 *	0.64	−2.44	3.43
Child: Negative affect	14.23 **	2.18	17.73	9.91	3.94 **	0.68	5.53	3.11	4.64 **	0.83	3.66	3.78
Mother: Sociability X Child: Negative affect			−0.95	2.62			−0.43	0.82			0.27	1.00
R-squared	0.27		0.27		0.22		0.22		0.29		0.29	

The associations are adjusted for age, gender, and adoption status. Values are regression coefficients (b) and standard errors (se). ** *p* < 0.01, * *p* < 0.05. CBCL: Child Behavior Checklist [37].

**Table 6 children-09-01227-t006:** The association between maternal psychological distress and child temperamental characteristics with child problem behavior.

	CBCL Total				CBCL Internal				CBCL External			
	**b**	**se**	**b**	**se**	**b**	**se**	**b**	**se**	**b**	**se**	**b**	**se**
Maternal psychological distress	1.30 **	0.47	−7.73 *	3.89	0.32 *	0.14	−2.13	1.17	0.46 *	0.18	−2.69	1.5
Child: Extraversion/Surgency	−4.74 *	2.29	−8.38 **	2.74	−2.65 **	0.69	−3.64 **	0.83	0.13	0.88	−1.14	1.06
Maternal psychological distress × Child: Extraversion/Surgency			1.93 *	0.82			0.52 *	0.25			0.67*	0.32
R-squared	0.13		0.16		0.16		0.18		0.16		0.18	
	**b**	**se**	**b**	**se**	**b**	**se**	**b**	**se**	**b**	**se**	**b**	**se**
Maternal psychological distress	0.92 *	0.44	1.07	2.92	0.21	0.14	−0.31	0.92	0.33	0.17	0.36	1.13
Child: Negative affect	13.23 **	2.19	13.30 **	2.55	3.73 **	0.69	3.49 **	0.81	4.38 **	0.85	4.39 **	0.99
Maternal psychological distress × Child: Negative affect			−0.05	0.83			0.15	0.26			−0.01	0.32
R-squared	0.27		0.27		0.22		0.22		0.28		0.28	

The associations are adjusted for age, gender, and adoption status. Values are regression coefficients (b) and standard errors (se). ** *p* < 0.01, * *p* < 0.05. CBCL: Child Behavior Checklist [37].

**Table 7 children-09-01227-t007:** The multivariate associations of child and maternal temperamental characteristics and maternal distress with child problem behaviors.

	CBCL Total		CBCL Internal		CBCL External	
	**b**	**se**	**b**	**se**	**b**	**se**
Mother: Negative affect	5.11	2.71	1.39	0.85	1.22	1.05
Child: Negative affect	11.38 **	2.31	2.87 **	0.72	4.37 **	0.89
Mother: Sociability	−2.19	1.69	−0.32	0.53	−1.61 *	0.65
Child: Extraversion/Surgency	−3.79	2.62	−	0.82	0.51	1.01
Maternal psychological distress	−7.02	3.55	−1.88	1.11	−2.69	1.37
Maternal psychological distress × Child: Extraversion/Surgency	1.61 *	0.76	0.43	0.24	0.61 *	0.29
R-squared	0.32		0.29		0.34	

Values are regression coefficients (b) and standard errors (se). ** *p* < 0.01, * *p* < 0.05.

## Data Availability

The data and the analytical code are available from the corresponding author upon reasonable request.

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
