# Peer review of "Do the Temperamental Characteristics of Both Mother and Child Influence the Well-Being of Adopted and Non-Adopted Children?"

_children, 2022, doi:10.3390/children9081227_

Round 1

Reviewer 1 Report

The article is well done. The subject is high to interest.

Author Response

Thank you for your review.

Reviewer 2 Report

The manuscript addresses interesting and relevant topic, which was especially popular several decades ago, e.g., goodness-of-fit between maternal and child temperamental characteristics. Importantly, these associations are analyzed in adopted and non-adopted children, as well as while linking these associations with the children problem behavior and taking into account the maternal distress. The limitations of the study – also clearly indicated by authors of the manuscript - are relatively small sample and maternal reported measures.

Several suggestions how to improve the manuscript:

The Abstract is too long, no need to present the name and contract no. of the Ethics committee, and the exact numbers (b, se, p) in it.

Keywords are not very relevant. The reference to "adoption" is presented trice (it is enough to leave one keyword, e.g. “internationally adopted children”); also, it is more relevant to specify maternal distress, and children behavioral problems instead of "psychological distress" and "mental wellbeing". This suggestion is optional.

The Introduction and Discussion should be more coherent and more sound. I strongly suggest to stay on the selected theoretical framework and / or background. As for example, recently, in Introduction the Extended evolutionary synthesis is presented, and the Discussion is based on the goodness-of-fit hypothesis regarding child-parent temperament and on a personality-resilience-psychopathology model. In addition, it is not clear why the authors introduce plenty of research and knowledge on the variables which explain the maternal depression. In the recent study, maternal distress is one of the variables of the study, not the object of the study. I would strongly recommend to clearly formulate and ground the main premises or hypothesis for this study in the Introduction.

Methods are clearly described, although the reliability metrics (Cronbach alphas) of the main scales for this study sample should be provided. Also, it would be very important to present more detailed characteristics of two study samples (adopted vs. non-adopted children) – if and how they are comparable in terms of children’s gender, age, maternal education? An inaccuracy while describing the CBCL is left in the first sentence (line 169): it is not 118-question checklist. Citations of (or references to) Hawk & McCall, 2010; Juffer, van Ijzendoorn, 2005; Verhulst et al., 1990; Biederman et al., 1993, 1995, 1996; Kazdin & Heidish, 1984; Weinstein et al., 1990; etc. are too redundant and outdated (there are more relevant and recent, although they are not necessary, as the CBCL is well-known instrument, it’s enough to refer to the original authors and translations to Finnish).

For child temperament and behavioral problems different forms of instruments (depending on child’s age) were used - whether the scores for the analysis then were standardized and how? It looks like the raw scores are presented in Table 1 and Table 2? Finally, it is not clear how the total CBCL score was calculated (adding up Internalizing and Externalizing score, or summing up the all items score)?

In Results, there is no need to repeat the same numbers (e.g., means, b, p, etc.) which are already presented in the tables. The same variable name should be used throughout the Results and Tables. Recently, for maternal distress, in Table 1 “psychological distress”, in Table 2 “GHQ”, in Table 3 “maternal distress”, in Figure 1 “maternal depression”! In addition, in Table 3 should be “CBCL” instead of “CCL”. It would be important to present the model fit characteristics and determination coefficients for the linear regression models.

The Discussion should be coherent and in match with the Introduction. Also it should consequently address the initial assumptions or hypothesis. Finally, as the main findings of the study are based on temperament characteristics (which are hardly modifiable) it would be very important to carefully and reasonably formulate the main recommendations for practitioners and / or for the interventions.

Author Response

Thank you for your review. Please see attachment.

The manuscript addresses interesting and relevant topic, which was especially popular several decades ago, e.g., goodness-of-fit between maternal and child temperamental characteristics. Importantly, these associations are analyzed in adopted and non-adopted children, as well as while linking these associations with the children problem behavior and taking into account the maternal distress. The limitations of the study – also clearly indicated by authors of the manuscript - are relatively small sample and maternal reported measures.

Several suggestions how to improve the manuscript:

The Abstract is too long, no need to present the name and contract no. of the Ethics committee, and the exact numbers (b, se, p) in it.

Keywords are not very relevant. The reference to "adoption" is presented trice (it is enough to leave one keyword, e.g. “internationally adopted children”); also, it is more relevant to specify maternal distress, and children behavioral problems instead of "psychological distress" and "mental wellbeing". This suggestion is optional.

The keywords have now been changed and the abstract has been shortened. The numerical values have been left out.

The Introduction and Discussion should be more coherent and more sound. I strongly suggest to stay on the selected theoretical framework and / or background. As for example, recently, in Introduction the Extended evolutionary synthesis is presented, and the Discussion is based on the goodness-of-fit hypothesis regarding child-parent temperament and on a personality-resilience-psychopathology model. In addition, it is not clear why the authors introduce plenty of research and knowledge on the variables which explain the maternal depression. In the recent study, maternal distress is one of the variables of the study, not the object of the study. I would strongly recommend to clearly formulate and ground the main premises or hypothesis for this study in the Introduction.

The introduction has been revised according to the comments and it is now more coherent in relation to the discussion.

Methods are clearly described, although the reliability metrics (Cronbach alphas) of the main scales for this study sample should be provided. Also, it would be very important to present more detailed characteristics of two study samples (adopted vs. non-adopted children) – if and how they are comparable in terms of children’s gender, age, maternal education? An inaccuracy while describing the CBCL is left in the first sentence (line 169): it is not 118-question checklist. Citations of (or references to) Hawk & McCall, 2010; Juffer, van Ijzendoorn, 2005; Verhulst et al., 1990; Biederman et al., 1993, 1995, 1996; Kazdin & Heidish, 1984; Weinstein et al., 1990; etc. are too redundant and outdated (there are more relevant and recent, although they are not necessary, as the CBCL is well-known instrument, it’s enough to refer to the original authors and translations to Finnish).

The description of the CBCL has  been clarified and shortened. Cronbach alphas have also now been reported.

For child temperament and behavioral problems different forms of instruments (depending on child’s age) were used - whether the scores for the analysis then were standardized and how? It looks like the raw scores are presented in Table 1 and Table 2? Finally, it is not clear how the total CBCL score was calculated (adding up Internalizing and Externalizing score, or summing up the all items score)?

Given that both CBQ and ECBQ have the same scale, we used unstandardized scores in the analyses. The total CBCL score have been calculated by summing up all items. We have now clarified these issues in the text.

In Results, there is no need to repeat the same numbers (e.g., means, b, p, etc.) which are already presented in the tables. The same variable name should be used throughout the Results and Tables. Recently, for maternal distress, in Table 1 “psychological distress”, in Table 2 “GHQ”, in Table 3 “maternal distress”, in Figure 1 “maternal depression”! In addition, in Table 3 should be “CBCL” instead of “CCL”. It would be important to present the model fit characteristics and determination coefficients for the linear regression models.

The numerical values have been removed and the variable names have been checked. We have also reported regression R2-values.

The Discussion should be coherent and in match with the Introduction. Also it should consequently address the initial assumptions or hypothesis. Finally, as the main findings of the study are based on temperament characteristics (which are hardly modifiable) it would be very important to carefully and reasonably formulate the main recommendations for practitioners and / or for the interventions.

The Discussion has been modified accordingly. Recommendations for practitioners have been included.

Reviewer 3 Report

Congratulations to the authors because the study is very interesting and the manuscript is well developed. The topic is very important and original. As far as I know, I haven't read too many studies controlling for adoptive parental psychological outcomes. The introduction is well written and allowes the reader to followed the logics of the study. However, I have some concerns about the statistical analyses. The authors has chosen the option of performing multiple regressions, when the study really requires one only big regression. It seems that the core results will remain the same, but we need to see the proper analyses to avoid spurious associations. I develop this issue and I propose other type of changes in the following comments: 

C1. Please, add the hypotheses of the study. After reading the introduction several hypothesis came to my mind, but we need to know which one was the main hypothesis.

C2. Why do not the authors explain anything about the sample in the participants subsection? Please, add sociodemographical information about the sample.  For example, the authors can add lines 210-211 to the participants section. But the descriptive statistics should remain at the beginning of the Results section.

C3. Please, calculate the reliability of the scales in your sample and report them. One test can be reliable in the validation study, but you need to demonstrate that the responses to that test in your sample are also reliable. The authors can choose adding this information in the Instruments section (it is where I recommend to include it) or in the Results section.

C4. Please, change Table 2. The p value is not so informative that the statistic. I supose you used a Student's T. So please, report the T. In addition, when we calculate differences between groups, we need to know the effect size. The effect size cannot be estimated through the Student's T. So, please, calculate it and interpret it properly. Regarding Table 2, please, clarify what means CBCL in Row 5. If it is a total score, please, add this information.

C5. Please, add points to any number inside Table 3. The standard way to report correlations is with decimals, not by percentatges. Instead of 100, the authors must write 1.00. Instead of -20, the authors must write -.20

C6. In lines 235 to 247, please add a subscript in every "b" reported to differentiate properly the results regarding maternal negative affectivity and child negative affectivity, for example.

C7. Please, the authors must be consistent in the way they refer to different concepts through the manuscript. Please, change "BP" in Table 4 for "CBCL" as the authors has used in previous tables.

C8. Hast he authors tried to perform a regression including Maternal psychological distress, Mother negative affectivity, Mother extraversion/Surgency, Child Negative affectivity and Child extraversion/surgency with each dependent variable? I encourage the authors to perform this analysis. It will be more informative than performing 5 different regression. With the whole sample, the authors has the statistical power enough to do that.

C9. It is a bit strange that the authors refers only to the effects of adoption vs non-adoption only in one line of the results section (L276). In addition, the tables in the supplementary material are not well explained to understand the results. In which sample has the authors performed the analyses of the supplementary material? Only in the adoption sample? So, I understand that the regressions analyses in the manuscript are performed with the whole sample. To better understand whether the authors found differences in the associations between independent and dependent variables between adopted and non-adopted children, show us the regression with both subsamples. The best way to test for these differences should be including the variable "adoption status" as an independent variable in the regression and test the interaction with all the other variables. In fact, according to the second study's objective, this one should be the statistical test performed. 

C10. Finally, the study lacks of an adequate comparision with past studies in the discussion section. The authors may also add some past studies into the introduction section. Temperament and personality are very related constructs. When we speak about temperament in adults, we must take into account personality studies too. So, please, look for studies that relates personality of parents and children. For example, see Xing et al., (2021; https://www.frontiersin.org/article/10.3389/fpsyg.2018.01952).

Round 2

Reviewer 2 Report

The authors have addressed most of the comments. The revision and edition of the manuscript are good and acceptable, although not perfect. I refrain from further comments on introduction and discussion, and admit that they are sufficiently relevant. 

Please, specify once more the calculation of CBCL total score. To the previous comment, that it is not clear how the total CBCL score was calculated, the authors responded, that "The total CBCL score have been calculated by summing up all items. We have now clarified these issues in the text." First, it was not clarified in the revised manuscript (lines 146-152). Secondly, if it is done as stated in the response, the mean score of CBCL Total cannot be less / lower than the mean score of CBCL Internalizing and / or CBCL Externalizing (see Table 1: M=9.41 for Total CBCL, and M=11.4 for Externalizing CBCL). Thus I guess, the miscalculation or inaccuracy in providing the results was left in the manuscript. Also, the authors provided in a revised manuscript the scale reliability only for CBCL total, but not for the Internalizing and Externalizing scales.

Finally, there are minor inaccuracies left while inserting the additions and editions of the manuscript (as for example, see lines 67-69).

Author Response

Thank you for the second review. Here is a point-by-point response to the comments.

Please, specify once more the calculation of CBCL total score. To the previous comment, that it is not clear how the total CBCL score was calculated, the authors responded, that "The total CBCL score have been calculated by summing up all items. We have now clarified these issues in the text." First, it was not clarified in the revised manuscript (lines 146-152). Secondly, if it is done as stated in the response, the mean score of CBCL Total cannot be less / lower than the mean score of CBCL Internalizing and / or CBCL Externalizing (see Table 1: M=9.41 for Total CBCL, and M=11.4 for Externalizing CBCL). Thus I guess, the miscalculation or inaccuracy in providing the results was left in the manuscript. Also, the authors provided in a revised manuscript the scale reliability only for CBCL total, but not for the Internalizing and Externalizing scales.

We apologize for this error and have now corrected it. Reliability coefficients have now also been provided to Internalizing and Externalizing scales.

Finally, there are minor inaccuracies left while inserting the additions and editions of the manuscript (as for example, see lines 67-69).

We have now gone through the manuscript and have corrected all inaccuracies.

Reviewer 3 Report

I express my gratitude to the authors for considering my comments. And I congratulate them because the manuscript has been improved. It is much clear now. However, it still lack of tiny but significant pieces of information and one great issue. Please, consider the following comments: 

C1. Please, add the standar deviation with the mean age in line 125. It is important to estimate the age range. Means always need standard deviations to be understable. I see you added the standard deviation to Table 1, but if you cite that mean in the text, you need also to report the SD.

C2. In Table 2, the t statistic has been reported. However, in the version I have reviewed, no Cohen's d can be seen. Please, make sure that the readers can see the Cohen's d.

C3. The authors explain that they tested for interactions in the last regression analyses. However, these results have not been displayed in Table 7. Please, add this information.

C4. We have arrived to the big issue. According to Table 7, there are several sentences in the discussion section that are not true. That is why the regression analyses reported in Table 7 were so necessary. In line 335, tha authors state that "Furthermore, maternal negative affectivity was associated with problem behaviors irrespective of child extraversion/surgency." This resault is based on the analysis reported in Table 4. However, when the authors have tested all the temperamental variables, including maternal and child temperament (Table 7), maternal temperament was not significant predicting any CBCL dependent variable. Only child temperament was a significant predictor. According with the results, the only relevant independent variables are the ones related to child temperament. So... I suggest that the authors re-write the discussion section according to the results.

C5. And... I'm afraid that we need some more additional analyses taking into account what we see in Table 7. We need the same analysis with the Maternal psychological distress. In fact, the ideal analysis should be adding Maternal psychologial distress to the same equations displayed in Table 7. If after controlling maternal and child temperament, Maternal psychological distress still predicts CBCL outcomes, then the authors can state that Maternal psychological distress is a robust predictor. If the authors do not perform this analysis, all the conclusions regarding Maternal psychological distress could be spurious. Please, re-write the discussino accordingly.

C6. According with comments 4 and 5... it seems that maternal temperament has a distant effect on CBCL outcomes. The correlation matrix show a strong correlation between maternal and child temperament. So... please, perform mediation analyses, because it seems that the effect of maternal temperament (and with a certain likelihood Maternal psychological distress) on CBCL outcomes is not explained by direct effects (these effects are tested with the regression analyses performed) but with indirect effects through temperament of children. This hypotheses is plausible because temperamental traits have a strong genetic load (as authors know). Try with path analyses or structural equation modelling or perform the classical steps developd by Baron and Kenny (1986).

References

Baron, R. M., & Kenny, D. A. (1986). The moderator-mediator variable distinction in social psychological research: conceptual, strategic, and statistical considerations. Journal of Personality and Social Psychology51(6), 1173–1182. Retrieved from http://www.ncbi.nlm.nih.gov/pubmed/3806354

Author Response

We would like to thank the reviewer for the feedback and for the additional suggestions. Here is a point-by-point response to the comments.

C1. Please, add the standar deviation with the mean age in line 125. It is important to estimate the age range. Means always need standard deviations to be understable. I see you added the standard deviation to Table 1, but if you cite that mean in the text, you need also to report the SD.

The SD has now been added to the text.

C2. In Table 2, the t statistic has been reported. However, in the version I have reviewed, no Cohen's d can be seen. Please, make sure that the readers can see the Cohen's d.

When the layout is horizontal, the Cohen’s d is visible.

C3. The authors explain that they tested for interactions in the last regression analyses. However, these results have not been displayed in Table 7. Please, add this information.

We have now included the Maternal psychological distress X Child: Extraversion/Surgency -interaction in the Table 7. The other potential interactions were not included as they were not statistically significant in the univariate models.

C4. We have arrived to the big issue. According to Table 7, there are several sentences in the discussion section that are not true. That is why the regression analyses reported in Table 7 were so necessary. In line 335, tha authors state that "Furthermore, maternal negative affectivity was associated with problem behaviors irrespective of child extraversion/surgency." This resault is based on the analysis reported in Table 4. However, when the authors have tested all the temperamental variables, including maternal and child temperament (Table 7), maternal temperament was not significant predicting any CBCL dependent variable. Only child temperament was a significant predictor. According with the results, the only relevant independent variables are the ones related to child temperament. So... I suggest that the authors re-write the discussion section according to the results.

We agree with the reviewer and have revised the manuscript according to the results presented in the Table 7.

C5. And... I'm afraid that we need some more additional analyses taking into account what we see in Table 7. We need the same analysis with the Maternal psychological distress. In fact, the ideal analysis should be adding Maternal psychologial distress to the same equations displayed in Table 7. If after controlling maternal and child temperament, Maternal psychological distress still predicts CBCL outcomes, then the authors can state that Maternal psychological distress is a robust predictor. If the authors do not perform this analysis, all the conclusions regarding Maternal psychological distress could be spurious. Please, re-write the discussion accordingly.

Maternal psychological distress has been included as predictor to the Table 7. We have revised the results and discussion accordingly.

C6. According with comments 4 and 5... it seems that maternal temperament has a distant effect on CBCL outcomes. The correlation matrix show a strong correlation between maternal and child temperament. So... please, perform mediation analyses, because it seems that the effect of maternal temperament (and with a certain likelihood Maternal psychological distress) on CBCL outcomes is not explained by direct effects (these effects are tested with the regression analyses performed) but with indirect effects through temperament of children. This hypotheses is plausible because temperamental traits have a strong genetic load (as authors know). Try with path analyses or structural equation modelling or perform the classical steps developd by Baron and Kenny (1986).

As suggested, we have now conducted additional mediation analyses. Results are as follows: 

Table. Unstandardized direct effects 

Model pathways 

SE 

% of total effect mediated 

Mother: Negative affectivity -> Child: Negative affectivity -> CBCL total 

3.11 

1.05 

2.95 

<0.01 

26 

Mother: Negative affectivity -> Child: Extraversion/Surgency -> CBCL total 

0.47 

0.57 

0.83 

0.41 

Mother: Sociability -> Child: Negative affectivity -> CBCL total 

-0.32 

0.70 

-0.46 

0.64 

11 

Mother: Sociability -> Child: Negative affectivity -> CBCL total 

-0.52 

0.40 

-1.31 

0.19 

17 

Mother: Psychological distress -> Child: Negative affectivity -> CBCL total 

0.24 

0.18 

1.36 

0.17 

19 

Mother: Psychological distress -> Extraversion/Surgency -> CBCL total 

-0.01 

0.06 

-0.10 

0.92 

As it can be seen from the table above, only child negativity affectivity mediates the effect of maternal negative affecitivity on child problem behaviors. We have included these results to the manuscript.